# The Global Financial Crisis and Overweight among Children of Single Parents: A Nationwide 10-Year Birth Cohort Study in Japan

**DOI:** 10.3390/ijerph16061001

**Published:** 2019-03-20

**Authors:** Koichiro Shiba, Naoki Kondo

**Affiliations:** 1Department of Social and Behavioral Sciences, Harvard T.H. Chan School of Public Health, Boston, MA 02115, USA; shiba_k@g.harvard.edu; 2Department of Health Education and Health Sociology, School of Public Health, The University of Tokyo, 7-3-1 Hongo, Bunkyo-ku, Tokyo 113-0033, Japan

**Keywords:** child overweight, recession, single parents, health disparity, social support

## Abstract

Evidence suggests that socioeconomically disadvantaged children may experience a greater increase in overweight risk during macroeconomic downturns. We examined whether inequalities in the risk of overweight between Japanese children from single- and two-parent households increased after the 2008 global financial crisis. We used data from ten waves (2001 to 2011) of a nationwide longitudinal survey following all Japanese children born within 2 weeks in 2001 (boys: *n* = 15,417, girls: *n* = 14,245). Child overweight was defined according to age- and sex-specific cut-offs for Body Mass Index (BMI). Interaction between a binary measure of crisis onset (September 2008) and single-parent status was assessed using generalized estimating equation models. Covariates included baseline household income and income loss during the crisis. Girls from single-parent households showed a greater increase in the odds of overweight after crisis onset (adjusted odds ratio (AOR), 1.23; 95% confidence interval (CI), 1.04–1.46) compared to girls from households with two parents, regardless of household financial status. A similar though statistically non-significant trend was observed among boys (AOR, 1.10; 95% CI, 0.92–1.30). Child overweight risk by single-parent status may increase during macroeconomic downturns, at least among girls. Financial aid to single-parent households may not suffice to redress this gap.

## 1. Introduction

Childhood overweight is a global public health challenge, linked both to childhood morbidity and increased risk of future chronic diseases [1,2,3]. In Japan, despite the relatively low prevalence of childhood overweight compared to Western, high-income countries, prevalence has increased in recent years [4,5]. In 2007, approximately 10% of 6-to-11-year-old children were overweight, representing a doubling in the childhood overweight prevalence since the late 1970s [4].

This increase in the prevalence of overweight is socially patterned [6,7]. Children from socially disadvantaged backgrounds face increased risk compared to their wealthier peers [8,9,10]. This gap may be growing. In Japan, a recent study using a nationwide, longitudinal data indicated a widening income-based gradient in overweight risk among children after the global financial crisis of 2008. During the recession, gross domestic product decreased by 6.3%, and the unemployment rate increased from 4% to 5.5% within a year [11] (pp. 158–174). People from lower socioeconomic backgrounds and workers with precarious employment status were more likely to suffer from these changes in macroeconomic conditions and resulting poverty [12].

Some pathways between social disadvantage and childhood overweight are well established. Poverty increases the risk of child overweight via inadequate access to nutritious food (i.e., food insecurity) [13,14,15]. Simultaneously, volatile macroeconomic conditions may arouse anxiety about the future and impose greater psychological stress on parents of socially disadvantaged households, which is also a risk factor for child overweight [14,16]. Both of these associations suggest that financial downturns should widen BMI inequalities by class. However, no study has fully explained observed widening inequalities by social disadvantage in the face of economic crises.

One under-examined explanation is family structure. Children from single-parent households may be disproportionately vulnerable to economic downturns, for two main reasons. First, because they have lower income, single-parent households are more likely to experience financial problems associated with changes in macroeconomic conditions (e.g., unemployment), resulting in food insecurity and other physical conditions that impose a higher risk. In Japan, in 2015, the relative poverty rate among households with only one adult was 50.8%, whereas it was 13.9% among households with two or more adults [17]. This poverty rate among single-parent households is the highest among all Organization for Economic Co-operation and Development member countries [18]. Moreover, nearly half of all single mothers, who constitute 90% of all single parents in Japan, do not have a full-time, secure job [19]. Secondly, single parents may experience a stronger psychological burden in response to financial shocks, as they lack instrumental and emotional social support in childrearing from their spouses. Such support ameliorates psychological distress via better stress coping [14,16,20]. Single parents may also tend to be socially isolated and lack sufficient social support because they tend to be busy working and taking care of their children, and so they may not have sufficient time to maintain social relationships.

Against this background, we hypothesized that the greater gradient in overweight risk between children from single-parent households and two-parent households widened after the 2008 global financial crisis. In addition, as a low degree of parental reception of social support is one of the rationales for potential vulnerability of children from single-parent households in terms of overweight risk, we hypothesized that parental reception of social support would buffer this widening, parenthood-based gradient in overweight risk. In this paper, we test this hypothesis by examining the relationship between single parenthood and trajectories of overweight risk before and during the financial crisis, using data from a nationwide, longitudinal birth cohort study. Children in this cohort were born in 2001 and were thus exposed to the economic downturn at a possibly sensitive age for the development of overweight [4,21].

## 2. Materials and Methods

### 2.1. Data

We used data from 10 waves (from 2001 to 2011) of the Longitudinal Survey of Newborns in the 21st Century (LSNC), conducted by the Ministry of Health, Labor, and Welfare in Japan [22]. The LSNC identified all babies born in Japan between January 10th and 17th and between July 10th and 17th in 2001 the birth record list of vital statistics for Japan (*N* = 53,575). Questionnaires were then mailed to the parents of these children at 0.5, 1.5, 2.5, 3.5, 4.5, 5.5, 7, 8, 9, and 10 years old. In total, 47,016 families participated in the survey (response rate: 88%) at baseline. For our analysis, we included only children with complete responses to the major variables (weight, height, income, and single parent status), eliminating 16,833 families. We further excluded 520 households who experienced a change in the number of parents in the same household from 2008 survey to 2009 survey, as we could not identify whether the change occurred before or after the global financial crisis onset (September 2008). Our final analytic sample thus included 29,662 families (See Figure 1).

### 2.2. Variables

For our outcome—binary overweight status—we categorized children based on their Body Mass Index (BMI), calculated as weight/height^2^ (kg/m^2^). At each wave, parents were asked to report children’s height (to the nearest 0.1 cm) and weight (to the nearest 0.1 kg), as well as a date of measurement. Childhood overweight was defined according to age- (in months, based on reported date of measurement) and gender-specific BMI cut-off points established by the International Obesity Task force, starting from two years of age [23]. These cut-off points project to adult cut-offs for overweight of 25 kg/m^2^.

To define our exposure of interest—single-parent household status—we used data on the number of parents (mother and father) cohabitating with a child in the same household, which were reported in 2008 (before the global financial crisis) and 2009 (after the global financial crisis). When both a father and a mother were living together in 2008 and 2009 continuously, we categorized children as living in a two-parent household. When only one parent was cohabitating in both 2008 and 2009, we categorized children as living in a single-parent household. We used the number of parents from the two waves to preclude the possibility of change in single-parent status around the global financial crisis.

Social support parents received from people other than their spouse during the global financial crisis was also measured. Specifically, parents were asked in a 2008 survey to answer whether they had someone to consult with when they have trouble with raising children and to select all sources of the support. Parents were categorized as receiving social support when they selected at least one of the non-spouse sources of social support including their parents, relatives, friends, teachers, counselors, other professionals or officers in public institutions, and websites.

Following a recent study using the same data, we treated household income before the global financial crisis, income reductions after the global financial crisis, parental educational attainments, parental age at birth, characteristics of residential areas, and cohabitation with their grandparent(s) as potential confounding factors [24]. Household income (defined as the sum of the father’s income, mother’s income, and other incomes) was reported by parents in 2001, 2002, 2004, 2005, 2006, 2007, and 2010. For our analysis, we equivalized household income by dividing the income by the square root of the number of household members. Household income before the global financial crisis was defined as an ordinal variable, dividing average annual equivalized household income into quartiles based on data from the 2002, 2005, and 2006 surveys (as the number of households members were only measured in these waves). To measure income reductions after the global financial crisis, we next created a binary variable representing a more than 30% income reduction comparing average household incomes between the waves of 2008 and 2010 (= 1 if income dropped by at least 30%, = 0 if it did not), following a recent report [24]. Parental educational attainments were measured at the 1.5-years of age follow-up. When at least one grandparent was cohabitating in a household, we categorized children as having grandparent(s) in the same household, as a study suggests cohabitation with grandparents is a potential risk factor for childhood overweight [25].

### 2.3. Statistical Analysis

First, we estimated trends in overweight prevalence before and after the global financial crisis onset among two-parent households and single-parent households, respectively.

Next, we evaluated whether changes in risk of child overweight after the onset of the global financial crisis differed with respect to single-parent status.

To do this, we constructed data based on 1-month intervals starting from January 2003, when the first group of children in the survey reached 2 years of age. We then used generalized estimating equation (GEE) models with an exchangeable correlation structure to calculate odds ratios (ORs) of overweight in each wave. Robust standard errors were used for hypothesis testing and confidence intervals. We used GEE in order to control for within-person correlation of errors across time points [26]. To offset the effect of temporal changes in overweight risks due to natural physical development, we also adjust for age (in months) in the regression models.

To assess the impact of the global financial crisis, we created a binary step term representing the data after crisis onset of September 2008 (before, = 0, or after, = 1) and then interaction terms between single-parent status and the crisis onset step term. Our sensitivity analysis using different step terms defined as varying 3-month intervals from June 2008 to December 2009 showed the model with the step term for September 2008 had the best model fit based on Bayesian information criterion (data available upon request). Using these explanatory variables, we formally tested, in Model 1, the difference in child overweight risk trajectories after the crisis onset between two-parent households and single-parent households. The coefficient for the interaction term between single-parent status and the step term indicates a difference in the post-crisis trajectory of the odds of overweight between single-parent households and two-parent households. In Model 2, we extend Model 1 by adjusting for household income (quartile) before 2008, negative income changes during the financial crisis, parental educational attainments, parental age at birth, characteristics of residential area, and cohabitation with grandparent(s) as covariates.

Finally, to see if the differential post-crisis trajectory in the odds of child overweight depending on the number of parents can be narrowed by social support, we re-ran Model 2 stratified by the presence of social support from non-spouse sources. Following suggested gender-specific characteristics on childhood overweight status, we stratified all models by gender [27]. All analyses were conducted using STATA SE statistical software, version 12.1 (Stata Corporation, College Station, TX, USA). The compute code used is available upon request. This research was based on secondary analysis of survey data that had already been anonymized unlinkably. Thus, no ethical review was required following the Ethical Guidelines for Medical and Health Research Involving Human Subjects.

## 3. Results

Of the 29,662 families who met our inclusion criteria and had full data, 15,417 of the children were boys and 14,245 were girls. Children living with both parents comprised 93.8% of boys and 93.5% of girls, and 6.2% of boys and 6.5% of girls were from single-parent households during the financial crisis (Table 1). Approximately 9% of both boys and girls experienced a 30% or more decrease in household income during the economic downturn. In regard to the trajectory of overweight prevalence for both genders, children from single-parent households showed a consistently higher prevalence of overweight compared to children from two-parent households. The gap in overweight prevalence between two-parent households and single-parent households appeared to widen after the financial crisis onset of September 2008 (Figure 2).

The results of our GEE models indicated there was a yearly increase in the odds of overweight among boys (OR: 1.02, 95% CI: 1.01–1.04) (Model 1 in Table 2). Further, for both genders, we observed a statistically significant increase in the odds of overweight after the financial crisis onset at September 2008 (OR: 1.62, 95% CI: 1.51–1.72 for boys and OR: 1.26, 95% CI: 1.18–1.35 for girls). For boys, the trend toward an increase in the odds of overweight after crisis onset was greater among children from single-parent households (OR: 1.10, 95% CI: 0.92–1.30), but this association was not statistically significant. For girls, in contrast, the post-crisis increase in overweight odds was greater among girls from single-parent households and statistically significant (OR: 1.23, 95% CI: 1.04–1.45). These findings remained constant even after additional adjustment of our covariates, including baseline household income levels and changes in income during the financial crisis (Model 2, see Appendix A for full results). A series of sensitivity analyses, including analyses that used different definitions of the onset of the financial crisis, alternative threshold values for percent income reductions during the crisis, and different income data for calculating average pre-crisis household income level, did not largely change the estimates or the final results (data available upon request).

Finally, when stratified by the presence of social support from non-spouse individuals and parent(s), the estimates for the interaction term between single-parent status and the step term differed across social support strata among both boys and girls (Appendix A). The excessive post-crisis increase in the odds of overweight among children from single-parent households was greater when their parents were not receiving non-spousal social support (OR, 1.25; 95% CI, 0.96–1.63 for boys and OR, 1.31; 95% CI, 1.00–1.71 for girls) compared to parents not receiving social support (OR, 1.01; 95% CI, 0.80–1.28 for boys and OR, 1.18; 95% CI, 0.94–1.48 for girls). We visualize these patterns in Figure 3, using predicted probabilities of overweight based on single parent status, presence of social support, and time period.

## 4. Discussion

The key findings of this study were twofold. First, children from single-parent households appeared to be at greater risk for developing overweight after the financial crisis, although it was statistically significant only among girls. These associations were independent of baseline household income levels and changes in household income level during the crisis period. Second, the excessive post-crisis increases in overweight risks among children from single-parent households appeared to be greater when their parent(s) did not have social support for their child care.

Although little study has examined the differential impact of macroeconomic downturn on overweight risk among children by their socioeconomic backgrounds, Ueda et al. found income-based widening disparity in child overweight risk after the financial crisis [24]. Our findings highlight the importance of socioeconomic circumstances other than income (i.e., family structure) in discussing the potential impact of macroeconomic condition on health disparities.

One hypothesis on the mechanism through which changes in macroeconomic conditions lead to child overweight is the resulting poverty that leads to food insecurity and less access to opportunities for physical activity [13,14,15]. However, we found a post-crisis widening disparity in overweight risk between single-parent and two-parent households even after adjusting for baseline household income levels and changes in income during the financial crisis. These data suggest that increased psychosocial stress among single parents due to the financial crisis is at least one important underlying cause.

In other words, while a sense of future insecurity due to the macroeconomic downturn may increase parental stress regardless of individual economic circumstances, single parents may also suffer from greater psychological distress, as they lack social support from a co-habiting partner. Social support that single parents could have received from their partner can be emotional (e.g., directly soothing stress by offering empathy and affection) or instrumental (e.g., cooperating and helping each other in childrearing), and numerous studies have shown both types of support to have protective effects against psychological distress [20,28]. Parental stress can increase child overweight risk because parents experiencing excessive stress may provide less supervision over children’s dietary and activity choices [13,29,30,31]. Furthermore, parents under stress may increase their reliance on precooked, calorie-dense meals and fast food to save time and reduce the effort required for meal preparation [29]. In addition, stress may reduce self-control or willpower by impeding parents’ cognitive function, leading to unhealthy dietary behavior and physical activity that may spill over to their children [32,33]. Moreover, spillover stress experienced by children can also increase the risk of overweight via cortisol release, which accelerates abdominal fat accumulation [32,34]. Though our analysis could not pick apart the reason single-parent status might lead to increased overweight risk among children, our secondary analysis showed that the excessive increase in child overweight risk among single-parent households is lower when parent(s) have more sources of social support, indicating social support can compensate for lack of spousal support.

It is worth noting that the differential trajectory of child overweight risks during and after the financial crisis by parenthood status was statistically significant only among girls. Although the statistical non-significant finding among boys does not mean absence of the association, it is possible that the adverse impact of macroeconomic downturns among children from single-parent households may differ by gender [27,35]. For instance, girls under psychological stress may consume more calorie-dense foods than boys do and be more susceptible to weight gain [36].

To our knowledge, this is the first study to evaluate the association between parenthood status and the trajectory of child overweight risks during and after the recent economic downturn. The strengths of our study include the use of large-scale, nationally representative, 10-year longitudinal data, with repeated measures of both our exposures and outcomes. However, several limitations should be noted when interpreting our findings. First, we did not directly evaluate the impacts of the financial crisis on the trajectory of overweight risks, but used the step term for the crisis onset, which may not fully represent how the crisis impacted the Japanese economy over time. Our previous sensitivity analysis using different cut-offs for the step term indicated that the model using September 2008 as our cut-off, which was the exact time of the crisis onset, had the best model fit [24]. This suggests that our observations of widening disparities in the risk of overweight during this time period are related to the financial crisis. Nevertheless, there may be other events that occurred exactly in September 2008 that explain the observed trends.

Second, some of our variables likely contain measurement error. Children’s weight and height were based on parental reports and therefore may not be precisely measured. However, parental reports of children’s weight status in Japan have been reported to be sufficiently precise [37]. Allowing parents to specify any date of measurement may also have increased the accuracy of parental reports. Similarly, our social support variable was measured by asking presence of someone to consult with when subjects have trouble with raising children and, thus, we could not identify what type of social support was offered, nor may we have captured every important aspect of social support.

## 5. Conclusions

Our research demonstrates that children from single-parent households experienced an excessive increase in overweight risk after the 2008 financial crisis, but that this parenthood-based difference in the impact of the crisis on child overweight risk was attenuated when single parents had social support from non-spousal sources. These findings suggest that the reason for widening inequalities in overweight after the financial crisis may be partially due to parental stress.

Despite the established universal health insurance coverage in Japan, single-parent households may be in further need of care. In addition to financial supports, more childrearing supports are needed to provide opportunities for single parents seeking jobs, social relationships, and support in raising their children.

## Figures and Tables

**Figure 1 ijerph-16-01001-f001:**
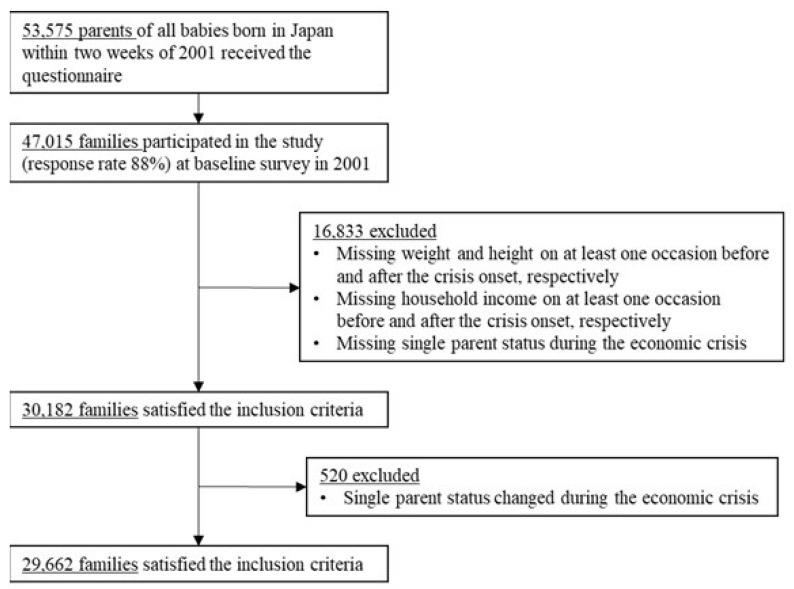
Participants flow for analytic sample (*n* = 29,662).

**Figure 2 ijerph-16-01001-f002:**
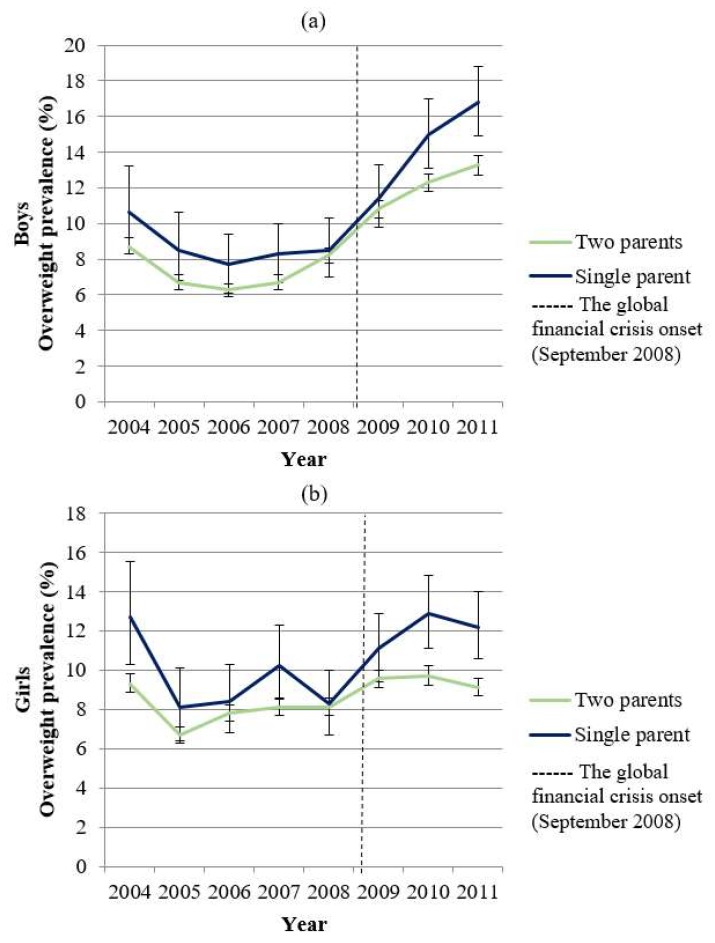
(**a**) Trajectory of overweight prevalence among boys by single parent status; (**b**) Trajectory of overweight prevalence among girls by single parent status. Error bars represent 95% confidence intervals.

**Figure 3 ijerph-16-01001-f003:**
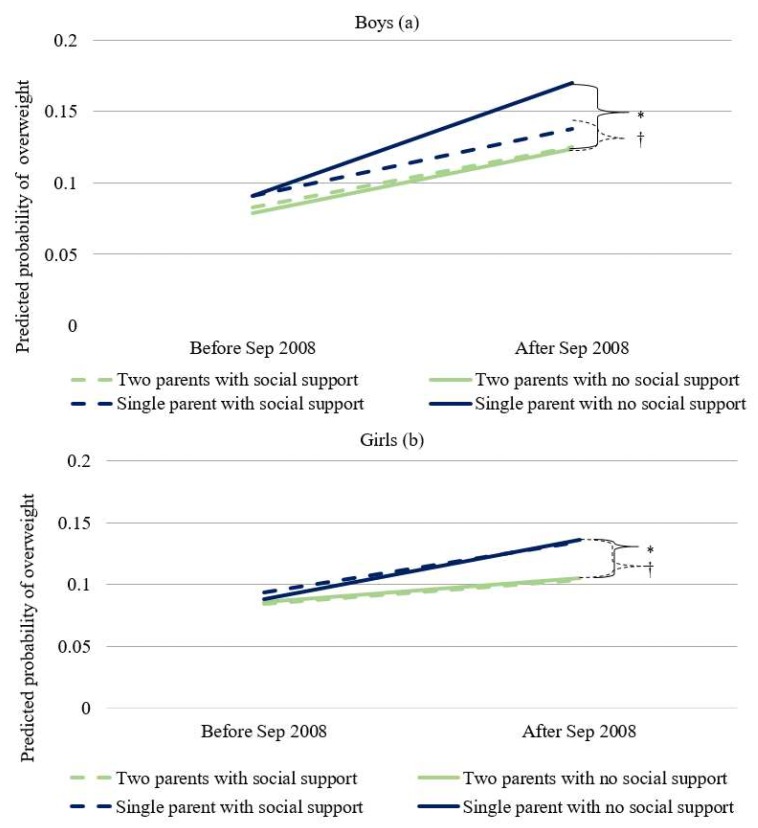
Difference in predicted overweight prevalence before and after the 2008 global financial crisis by parental status and parental reception of social support (**a**) among boys and (**b**) among girls. * *p*-Value for slope difference (with no social support): boys = 0.098, girls = 0.049, † *p*-Value for slope difference (with social support): boys = 0.908, girls = 0.140.

**Table 1 ijerph-16-01001-t001:** Demographic characteristics of the analyzed subjects by gender.

Characteristic	Boys		Girls	
	*n*	%	*n*	%
Total	15,417	52.0	14,245	48.0
Number of parents during the global financial crisis				
Two parents	14,461	93.8	13,317	93.5
Single parent	956	6.2	928	6.5
Household income quartiles before the global financial crisis				
1 (lowest)	3805	24.7	3497	24.5
2	3848	25.0	3578	25.1
3	3894	25.3	3584	25.2
4 (highest)	3870	25.1	3586	25.2
>30% negative income change during the global financial crisis	1355	8.8	1289	9.0
Mother’s education				
Junior high school	382	2.5	353	2.5
High school	5726	37.8	5362	38.2
Vocational school	6705	44.2	6064	43.2
Higher education	2352	15.5	2243	16.0
Missing	252		223	
Father’s education				
Junior high school	782	5.2	711	5.1
High school	5826	38.7	5416	39.0
Vocational school	2436	16.2	2200	15.8
Higher education	6012	39.9	5577	40.1
Missing	361		341	
Mother’s age at birth in 2001				
<20 years	98	0.6	72	0.5
21–25 years	1324	8.6	1294	9.1
26–30 years	5952	38.6	5407	38.0
>30 years	8043	52.2	7472	52.5
Father’s age at birth in 2001				
<20 years	41	0.3	38	0.3
21–25 years	833	5.4	815	5.8
26–30 years	4275	27.9	3926	27.8
>30 years	10,152	66.3	9357	66.2
Missing	116		109	
Residential area				
20 designated cities	3964	25.8	3635	25.6
Other cities	9992	65.0	9276	65.3
Rural	1416	9.2	1289	9.1
Missing	45		45	
Cohabitation with grandparents	3548	23.0	3173	22.3
Social support from non-spousal individuals	7953	51.6	7197	50.5

**Table 2 ijerph-16-01001-t002:** Odds ratio (OR) and 95% confidence interval (CI) for risk of overweight relative to normal weight by gender.

Variables	Model 1				Model 2			
	Boys		Girls		Boys		Girls	
	OR	(95% CI)	OR	(95% CI)	OR	(95% CI)	OR	(95% CI)
Age (years)	1.02	(1.01–1.04)	0.99	(0.97–1.00)	1.02	(1.01–1.04)	0.99	(0.97–1.00)
Parenthood								
Two parents	1.00 (Ref.)		1.00 (Ref.)		1.00 (Ref.)		1.00 (Ref.)	
Single parent	1.14	(0.97–1.33)	1.09	(0.93–1.28)	0.97	(0.81–1.15)	0.94	(0.79–1.12)
Step term								
Before September 2008	1.00 (Ref.)		1.00 (Ref.)		1.00 (Ref.)		1.00 (Ref.)	
After September 2008	1.62	(1.51–1.72)	1.26	(1.18–1.35)	1.62	(1.51–1.73)	1.26	(1.18–1.35)
Interaction between single parent in 2008 and step term								
Two parents * September 2008	1.00 (Ref.)		1.00 (Ref.)		1.00 (Ref.)		1.00 (Ref.)	
Single parent * September 2008	1.10	(0.92–1.30)	1.23	(1.04–1.45)	1.10	(0.93–1.31)	1.23	(1.04–1.46)

A generalized estimating equation model with an exchangeable correlation structure was used for the analysis. Robust standard errors were used to calculate 95% confidence intervals. Model 2 was adjusted for household income quartile before 2008, onset of 30% or more negative income change during economic crisis, mother’s education, father’s education, mother’s age at birth, father’s age at birth, residential area, and three-generation households. “Ref.” indicates a reference group. We used “*” to denote a product term between parenthood status and the step term.

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
