# Peer review of "The Global Financial Crisis and Overweight among Children of Single Parents: A Nationwide 10-Year Birth Cohort Study in Japan"

_ijerph, 2019, doi:10.3390/ijerph16061001_

Round 1

Reviewer 1 Report

Paper Title

There are very few papers that talk about childhood overweight. It is phrase that doesn't particularly sound correct.   I suggest that you either change it to Childhood Obesity or Children who are overweight but I leave the decision up to you. 

Abstract

This is written fairly well.

Line 16 Cut-offs not cutoffs.

1. Introduction

Please see my comment on the Title. 

Line 45 Try to avoid using phrases such as existing research without giving some examples.  

Line 48 are should be have and Line 49 a semi colon is required after income;

2. Materials and Methods

Is there a reference for the longitudinal data that you used?

Figure 1 is clear.  

Line 89 BMI cut-off points. These cut-off points.

3. Results

Table 1 is clear

Table 2 is clear

Figure 2 The two lines are barely indistinguishable from each other. You need to make one much darker than the other line in both graphs. As they currently stand, they don't make sense. 

Figure 3 This is better but one of the lines could be slightly darker than the other line in both graphs.

4. Discussion

I would like to see more comparison with existing literature in this section. 

Line 237 cut-offs

Author Response

We thank the reviewer for the fruitful comments. As below, we provide point-by-point responses to the reviewer's comments.

Reviewer 1

Paper Title

There are very few papers that talk about childhood overweight. It is phrase that doesn't particularly sound correct.   I suggest that you either change it to Childhood Obesity or Children who are overweight but I leave the decision up to you.

Overweight, typically defined as BMI of 25 or greater among adult populations, is different from obesity, which is defined as BMI of 30 or greater. Due to the distinction between overweight and obesity, we decided not to use the phrase childhood obesity. We believe the phrase “children among overweight” in the title is acceptable.

Abstract

Line 16 Cut-offs not cutoffs.

Corrected.

1. Introduction

Please see my comment on the Title.

Line 45 Try to avoid using phrases such as existing research without giving some examples. 

The purpose of this sentence is not to refer to particular published papers but instead to mention general paucity of knowledge with regard to the widening inequalities in child overweight risk during economic crises. Thus, we edited the text as follows.

Original sentence:

“However, existing research has not fully explained observed widening inequalities by social disadvantage in the face of economic crises.”

Revised sentence:

 “However, no study has fully explained observed widening inequalities by social disadvantage in the face of economic crises.”

Line 48 are should be have and Line 49 a semi colon is required after income;

We believe that comma, instead of semi colon, should be used in Line 49 after the word “income”. We have confirmed this with multiple native English speakers.

2. Materials and Methods

Is there a reference for the longitudinal data that you used?

We cited the following paper in this sentence.

Kana Fuse, Nobuo Nishi, Nayu Ikeda; Cohort Profile: 2001 Cohort of the Longitudinal Survey of Newborns in the 21st Century, International Journal of Epidemiology, Volume 46, Issue 5, 1 October 2017, Pages 1398–1398f, https://doi.org/10.1093/ije/dyx104

Line 89 BMI cut-off points. These cut-off points.

Corrected.

3. Results

Table 1 is clear

Table 2 is clear

Figure 2 The two lines are barely indistinguishable from each other. You need to make one much darker than the other line in both graphs. As they currently stand, they don't make sense.

We changed the colors of the lines in the Figure 2 so the contrast is clear even when it is printed only with black and white ink.

Figure 3 This is better but one of the lines could be slightly darker than the other line in both graphs.

We changed the colors of the lines in the Figure 3 so the contrast is clear even when it is printed only with black and white ink. We used the same colors as the ones used in the revised Figure 2.

4. Discussion

I would like to see more comparison with existing literature in this section.

Following the reviewer’s comment we added the following text discussing comparison of our results with existing literature.

“Although little study has examined the differential impact of macroeconomic downturn on overweight risk among children by their socioeconomic backgrounds, Ueda et al found income-based widening disparity in child overweight risk after the financial crisis [24]. Our findings highlight the importance of socioeconomic circumstances other than income (i.e. family structure) in discussing the potential impact of macroeconomic condition on health disparities.”

Line 237 cut-offs

Corrected.

Reviewer 2 Report

This manuscript addresses an important epidemiological topic, which is the impact of financial conditions on body composition, particularly in children. 

The study is well designed and conducted, data is clearly presented and the conclusions are appropriate. 

The findings of this study will be important to enacting adequate interventions to prevent childhood overweight and obesity, which are well established risk factors for metabolic disorders in adulthood. 

I would only recommend for the authors to address a few minor comments:

1- The financial crisis seemed to have had a greater impact in girls than in boys households ( 30% vs 9%). Despite there is no way from the research protocol to estimate the reasons for this fact, it could have had an impact on body composition differences between genders in children. In particular, the fact that financial support was proven insufficient to arrest the negative impact in girls body composition, but not in boys. This should be addressed in the discussion as an alternative explanation. 

2- Given the previous comment, I would recommend to rephrase the last sentence of the abstract accordingly. 

3- Authors should use the wording "non-significant" instead of insignificant when expressing statistical significance. 

Author Response

We thank the reviewer for the fruitful comments. As below, we provide point-by-point responses to the reviewer's comments.

Reviewer 2

This manuscript addresses an important epidemiological topic, which is the impact of financial conditions on body composition, particularly in children.

The study is well designed and conducted, data is clearly presented and the conclusions are appropriate.

The findings of this study will be important to enacting adequate interventions to prevent childhood overweight and obesity, which are well established risk factors for metabolic disorders in adulthood.

I would only recommend for the authors to address a few minor comments:

1- The financial crisis seemed to have had a greater impact in girls than in boys households ( 30% vs 9%). Despite there is no way from the research protocol to estimate the reasons for this fact, it could have had an impact on body composition differences between genders in children. In particular, the fact that financial support was proven insufficient to arrest the negative impact in girls body composition, but not in boys. This should be addressed in the discussion as an alternative explanation.

Following the reviewer’s comment on the seemingly gender-specific impact of the financial crisis, we added the following text in the Discussion.

“It is worth noting that the differential trajectory of child overweight risks during and after the financial crisis by parenthood status was statistically significant only among girls. Although the statistical non-significant finding among boys does not mean absence of the association, it is possible that the adverse impact of macroeconomic downturns among children from single-parent households may differ by gender [27,35]. For instance, girls under psychological stress may consume more calorie-dense foods than boys do and be more susceptible to weight gain [36].”

For the reviewer’s comment on the possibility that financial status plays different roles by gender, we believe that the association between single parenthood and post-crisis increase in overweight risk is equally independent of financial status for both genders. The point estimate and its confidence interval did not change after adjusting for financial status in both genders. Thus, we decided not to add the text discussing this point.

2- Given the previous comment, I would recommend to rephrase the last sentence of the abstract accordingly.

Given the possibility of gender-specific impact of the financial crisis on overweight risk, we modified the sentence in the abstract as follows:

Original:

“Disparities in child overweight risk by single-parent status may widen during macroeconomic downturns.”

Revised:

“Disparities in child overweight risk by single-parent status may widen during macroeconomic downturns, at least among girls.”

3- Authors should use the wording "non-significant" instead of insignificant when expressing statistical significance.

Corrected.